# The Association between Menstrual Symptoms and Presenteeism: A Cross-Sectional Study for Women Working in Central Tokyo

**DOI:** 10.3390/ijerph21030313

**Published:** 2024-03-08

**Authors:** Masumi Okamoto, Kumi Matsumura, Akiko Takahashi, Akio Kurokawa, Yuko Watanabe, Hiroto Narimatsu, Honami Yoshida

**Affiliations:** 1Center for Innovation Policy, Kanagawa University of Human Services, Kawasaki 210-0821, Japan; m.okamoto-mw7@kuhs.ac.jp (M.O.); a.kurokawa-4k5@kuhs.ac.jp (A.K.); y.watanabe-g8m@kuhs.ac.jp (Y.W.); h.narimatsu-3ry@kuhs.ac.jp (H.N.); 2Kanagawa Prefectural Government, Yokohama 231-0021, Japan; arakawa.3nj6@pref.kanagawa.lg.jp; 3School of Health Innovation, Kanagawa University of Human Services, Kawasaki 210-0821, Japan; 4Cancer Prevention and Control Division, Kanagawa Cancer Center Research Institute, Yokohama 241-8515, Japan; 5Department of Genetic Medicine, Kanagawa Cancer Center, Yokohama 241-8515, Japan

**Keywords:** women’s health, occupational health, working women, work productivity, work performance, presenteeism, menstruation

## Abstract

Menstrual symptoms lower women’s work performance, but to what extent one’s performance declines during the perimenstrual periods is unclear. This cross-sectional study evaluated relative presenteeism by the severity of menstrual symptoms in working women. Participants included women who joined a health promotion event in Tokyo. The severity of PMS and symptoms during menstruation were categorized based on their frequency, and the outcome variable was relative presenteeism as the ratio of work performance during the perimenstrual periods to that during the inter-menstrual period. An analysis of variance (ANOVA) was performed. Of the 312 participants, 238 were eligible, 50% of whom claimed severe symptoms in either PMS or during menstruation. Participants were divided into four groups (1) without severe menstrual symptoms, (2) severe PMS alone, (3) severe symptoms during menstruation alone, and (4) both severe PMS and symptoms during menstruation—and the mean relative presenteeism was 91% (standard deviation (SD) 23), 69% (SD 21), 76% (SD 16), and 69% (SD 27), respectively (*p* < 0.01). A between-group comparison revealed statistically significant differences in relative presenteeism, when group (1) served as the criterion for comparisons (*p* < 0.01). This study demonstrates that severe PMS alone, as well as both severe PMS and symptoms during menstruation, particularly decreased work performance.

## 1. Introduction

Menstrual symptoms, such as premenstrual syndrome (PMS), menstrual pain, and heavy menstrual bleeding, considerably lower women’s health-related quality of life [1,2,3,4,5]. PMS is described as either a poor psychological condition, such as irritability and anxiety, or physical discomfort, such as bloating, daytime drowsiness, and increased or decreased appetite, which typically lasts between 7 and 14 days before menstruation and disappears after it begins [6]. During menstruation, 45–90% of women experience menses-associated health problems, including menstrual pain, heavy menstrual bleeding, anxiety, and depression [7]. In severe cases, menstrual symptoms continue for more than two weeks, and menstrual pain and excessive blood loss, in particular, likely result in a more significant impact on women’s quality of life.

The impact of menstrual symptoms on work performance, such as absenteeism (being absent from work), job performance, and productivity, has gained attention in business settings and its significance has been examined [4,8,9,10,11,12]. One study in the Netherlands found that 13.8% of 32,748 women reported absenteeism during their menstrual periods, with 3.4% reporting absenteeism every or almost every menstrual cycle [13]. Studies targeting Japanese adolescents revealed a similar trend, with approximately 11% of students being absent due to PMS [14]. In addition to absenteeism, the impact of presenteeism (working while sick) in a state of low productivity, such as reduced work volume and working hours due to menstrual symptoms, is particularly intense [15]. This is because while presenteeism is sometimes invisible and difficult to evaluate, it affects corporate performance and the economy. In the abovementioned study in the Netherlands [13], 84% of total respondents (*n* = 26,438) reported presenteeism and productivity loss with a mean of 23.2 days per year. From an economic point of view, the estimated annual indirect costs due to decreased productivity associated with menstrual symptoms are USD 4,333 per person per year in the US [16] and JPY 491.1 billion per year in Japan as a whole [17]. Considering that those who are unable to continue working, have given up on promotions, or quit their jobs due to menstrual-associated health problems were not taken into account, the adverse impact of menstrual symptoms is expected to be even more severe.

Most previous studies have separately examined the relationship between the presence or severity of menstrual symptoms and work performance in PMS or menses-associated symptoms [3,8,18,19]. This may be because the mechanisms, causes, and coping strategies for PMS and its symptoms during menstruation are not always the same. Few studies have quantitatively assessed the extent to which intraindividual performance changes between the premenstrual and menstrual periods altogether. Furthermore, the extent to which there are interindividual differences in performance changes within the menstrual cycle depending on the presence or absence of severe menstrual symptoms is unclear.

There is no quantitative method to evaluate presenteeism in the context of menstrual symptoms with academically verified reliability or validity. In previous studies on menstrual symptoms and performance, the World Health Organization—Health and Work Performance Questionnaire (WHO-HPQ) [20,21], the Stanford Presenteeism Scale [22], and the Work Productivity and Activity Impairment Questionnaire [23] were commonly used to assess performance during menstrual periods on a 10-point scale ranging from “worst possible work performance” to “best work performance” [24,25,26]. In most studies, only the performance during the presence of menstrual symptoms (absolute absenteeism) [24], the total number of days of perceived decrease in workload and working hours [17], or the proportion of those who experienced a decrease in performance during the period when menstrual symptoms were present [19] had been calculated. However, no comparison was made with the state of performance during the menstrual cycle.

In order to address this issue, the concept of relative presenteeism [27], which is calculated from the ratio of the performance of an individual to that of other workers performing the same job (possible performance) and is commonly used in the fields of management and occupational health [25,28,29], may be useful. Contrary to absolute presenteeism just rating on a scale ranging from “worst possible work performance” to “best work performance, relative presenteeism helps us to more accurately understand the extent to which one’s performance declines compared to their possible performance. In light of this, we anticipate that incorporating the concept of relative presenteeism into menstrual symptom studies can enable us to examine the degree of change in work performance during the presence of menstrual symptoms, compared with the inter-menstrual period (periods without any symptoms).

Our study hypothesizes that presenteeism during perimenstrual periods is lower than that during periods without any menstrual symptoms, and that relative presenteeism is much lower for those who experience both severe PMS and menses-associated health problems than those who have no or slight PMS or menses-associated problems.

This study aimed to evaluate relative presenteeism as the ratio of work performance during the perimenstrual periods to that during inter-menstrual periods by the severity of menstrual symptoms in women working in central Tokyo. We decided to focus on working women in central Tokyo, as this was a preliminary study and the scope of our work is to expand nationwide in the future, with this study’s findings expected to comprise fundamental information. Moreover, approximately 3.90 million women are employed [30] and 84.6% are employed in the tertiary industry [31] in Tokyo. Therefore, the sociodemographic characteristics of the participants are relatively homogeneous, thereby making it possible to maintain high comparability.

## 2. Methods

### 2.1. Study Design

This was a cross-sectional study, and the survey was conducted in Japanese during a health promotion event, called Marunouchi Hokenshitsu (“The infirmary in Marunouchi area”, in Japanese), between 2 and 16 October 2021 in the business district in Chiyoda Wards, Tokyo. This event took place under the framework of the “Will Conscious Marunouchi” project, a platform for promoting working women’s health, initiated by the Mitsubishi Estate Company, Limited (Tokyo, Japan). This study was conducted in collaboration with the Mitsubishi Estate Company, Limited, Femmes Médicaux, Co. (Tokyo, Japan) and the Kanagawa University of Human Services. All data in this study were collected during the aforementioned health promotion event, and anonymized data were used for analysis.

### 2.2. Procedures and Population

Nine companies in the Marunouchi area joined through an invitation from the Mitsubishi Estate Company, Limited, asking their employees to participate in the event. Moreover, the Mitsubishi Estate Company, Limited, publicized the event through press releases and its website to call individuals to participate. The fee for participants via companies was free. An explanation of this survey was provided online, and informed consent by the opt-out method was obtained from all participants. Once consent was obtained, all participants completed a self-rated questionnaire online and underwent physical health evaluations, including blood tests and gynecological examinations. The respondents who did not understand Japanese, were not employed, were absent from work in the past month, were medicated for mental illness, were pregnant or possibly pregnant, were menopausal, or were taking pills or other hormonal drugs were excluded from this study. The participants were not restricted to any nationality or country of origin. The final participants comprised 312 women. As the announcement of this study was publicized on intranets in the abovementioned nine companies and to an unspecified number of people through press releases and posts on the website, the exact study population remained unclear.

### 2.3. Measurements

#### 2.3.1. Menstrual Symptoms

Participants were asked regarding the frequency of the following menstrual symptoms: irregular and unpredictable menstrual cycles, missed periods of over three months, short periods (five days or fewer), excessive bleeding and long periods (more than eight days), and severe menstrual pain, including lower abdominal or lower back pain. This study also asked the participants whether they were currently experiencing PMS. The symptoms of PMS were categorized as physical discomfort (bloating, nausea, fatigue, daytime drowsiness, increased or decreased appetite, head or lower back pain, diarrhea, etc.) and mental discomfort (irritability, emotional or feeling sad, etc.). Both the frequency of menstrual symptoms and physical and mental discomfort in PMS were evaluated separately and rated using a four-point scale (“no symptoms”, “rarely”, “sometimes”, “always”).

Participants were categorized into four groups: (1) the group without severe PMS or menstrual symptoms, if they reported “no symptoms”, “rarely”, and “sometimes” for all menstrual symptoms; (2) the group with any of the symptoms of severe PMS alone, if they answered “always” for either premenstrual physical or mental discomforts without answering “always” for symptoms during menstruation; (3) the group with any of the severe symptoms during menstruation alone, if they answered “always” for any symptoms during menstruation without answering “always” for PMS; and (4) the group with both severe PMS and symptoms during menstruation, if they reported “always” for both PMS and any symptoms during menstruation.

#### 2.3.2. Relative Presenteeism

Relative presenteeism is the ratio of actual performance to the work performance of workers in the same job (possible performance) [25,29]. The work performance of relative presenteeism is assessed by the WHO-HPQ with a 10-point scale ranging from “worst possible work performance” to “best work performance”. A higher ratio indicates a lower lost performance, meaning that a higher ratio indicates higher performance [27]. In this study, we modified this scale to calculate the ratio of performance during the perimenstrual period to that during the inter-menstrual period to assess the extent of change in work performance occurring throughout participants’ menstrual cycles.

### 2.4. Covariates

Covariates were selected based on previous studies [17,18,24]. The variables related to the participants’ attributes included age, academic background (university graduate, other), marital status (married or unmarried), and the presence of children, all of which were used as covariates in another study using the same datasets [32]. Work-related factors such as work-life balance and perceived work demands were reported to be associated with work performance [18]. Similarly, this study used work-related factors from the Job Stress Questionnaire [33] (friendly atmosphere in the workplace, work engagement, job self-control, and work-self balance), as well as some original items (easy leave, flexi-time or short-working-hour system, and overworking hours per day). Friendly atmosphere in workplace (“The atmosphere in my workplace is friendly”) and work engagement (“I feel invigorated when I am working”) were treated as dichotomous variables: (1) “Yes”, if they reported “very much so” or “moderately so”, and (2) “No”, if they reported “somewhat” or “not at all.” The job self-control score was the sum of three questions rated on a four-point scale (“I can work at my own pace”, “I can choose how and in what order to do my work”, and “I can reflect my opinions on workplace policy”) with higher scores indicating a higher degree of self-control. The score of work-self balance was the sum of a four-point scale (“very much so”, “moderately so”, “somewhat”, and “not at all”) on four questions: two for positive and another two for negative. Positive work-self balance is defined as being energized by work and using skills and knowledge obtained at work to enhance life, while negative work-self balance is defined as recognizing that one’s personal life suffers because of thinking only about work and its schedule. We positively reversed the responses to the questions on work-self balance; thus, higher scores indicated higher work-self balance. Regarding health-related factors, general health, the Kessler Psychological Distress Scale (K6) [34], knowledge of menstrual symptoms, and consultations on women’s health issues were used. Although other factors such as stress have been recognized as effecting the severity of menstrual symptoms among Japanese women [35,36], its effect was not adjusted as it was not included in the questionnaire. General health was assessed on five-point scale, with a higher score indicating lower subjective health. K6 is six-item screening scale with a five-point rating system that evaluates psychological distress in the preceding month. Higher scores indicate higher levels of psychological distress. Knowledge of PMS and symptoms during menstruation was evaluated on a four-point scale, and participants were divided into two groups: (1) “no knowledge of symptoms” if they answered “do not know”, “heard of the name”, or “know the symptoms but not how to deal with it”, and (2) “knowledge of symptoms” if they answered “know more about the symptoms and how to deal with it.” Similarly, for consultation with women’s health issues, participants were divided into two groups: (1) “No”, if participants had no one to discuss their health concerns with, such as health specialists, colleagues, friends, or family, and (2) “Yes”, if participants had anyone with whom they could discuss their health concerns.

### 2.5. Statistical Analyses

First, relative presenteeism was calculated as the ratio of performance during the perimenstrual period to that during the inter-menstrual period. An analysis of variance (ANOVA) was performed to confirm whether statistical differences in relative presenteeism existed among the four groups: (1) the group without severe PMS or menstrual symptom, (2) the group with any of the symptoms of severe PMS alone, (3) the group with any of the severe symptoms during menstruation alone, and (4) the group with both severe PMS and symptoms during menstruation. Subsequently, Tukey’s test was conducted to clarify the differences between every two of the four groups. A logistic regression analysis was complementarily conducted as a sensitivity analysis to enhance the results’ credibility. Although cutoff values of relative presenteeism were not scientifically verified in the field of menstrual health, this study adapted 80% of the reduction in presenteeism in accordance with evidence from the findings of an occupational health study [37]. Multiple regression analyses were conducted to demonstrate the association between the presence of severe menstrual symptoms and relative presenteeism in Model 1. Model 2 was adjusted for the following basic attributes: age, academic background, marital status, and the presence of children. Model 3 was adjusted for work-related factors: friendly atmosphere in the workplace, work engagement, job self-control, work-self balance, easy leave, easy use of flexi-time or a short-working-hour system, and overworking hours per day. Model 4 included health-related factors, including general health, K6, literacy of menstruation symptoms, and consultation on women’s health issues as covariates. To account for multicollinearity issues, we used the Pearson correlation coefficient for interval variables and a t-test for ordinal variables to assess the relationship of each candidate variable on menstrual symptoms and relative presenteeism. We also checked the variance inflation factor of all the variables used in the multi-regression analyses to ensure that they were less than 10. All statistical analyses were performed using the R software (version 4.2.2) (R Core Team, Vienna, Austria). 

This study was conducted in accordance with the Declaration of Helsinki and approved by the Institutional Review Board of the Kanagawa University of Human Services (approval number: SHI No.20).

## 3. Results

### 3.1. Participants’ Characteristics

In total, 312 working women in Marunouchi completed this study. Of them, 238 were eligible for this study; three were not employed, three were absent from work in the past month, eight were under medical treatment for mental illness, three were pregnant or possibly pregnant, fifty-one were taking pills or hormone drugs, seven were menopausal, and one had overlapping exclusion criteria. All age groups were included in the analysis, even if they were over 50 years old, except for those who reported menopause. Table 1 shows the characteristics of the participants. The age range of the patients was 20s–60s. The most common age group was participants in their 30s (39.9%), with a mean age of 35.8 (standard deviation [SD]: 8.0), and unmarried participants accounted for 60.5% of the total. Regarding menstrual symptoms, (1) the group without severe PMS or menstrual symptom comprised 119 (50.0%) participants, (2) the group with any of the symptoms of severe PMS alone comprised 50 (21.0%), (3) the group with any of the severe symptoms during menstruation alone comprised 23 (9.7%), and (4) the group with both severe PMS and symptoms during menstruation comprised 46 (19.3%). More than half of participants reported working at least one hour of overtime per day and 5.4% claimed over five hours of overtime working. As for general health, those who responded “Moderate” or “Not so bad” accounted for 32.4%, respectively, and the mean K6 score was 5.1 (SD: 4.9).

### 3.2. Work Performance by Presence or Absence of Menstrual Symptoms

Table 2 displays the work performance of the four groups. The mean relative presenteeism was 81% (SD: 25) as a whole and the figures for each category were 91% (SD: 23), 69% (SD: 21), 76% (SD: 16), and 69% (SD: 27), respectively. In the comparison between groups, the difference in relative presenteeism was −22 (*p* < 0.01) in group (2), −16 (*p* = 0.02) in group (3), and −23 (*p* < 0.01) in group (4), when group (1) served as the criterion for comparisons. Likewise, the sensibility analysis with logistic regression analysis resulted in similar findings (Appendix A).

### 3.3. The Association between Menstrual Symptoms and Work Performance

Table 3 presents the results of the multi-regression analysis. After adjusting for demographic, work-related, and health-related factors, the group with any of the symptoms of severe PMS alone (regression coefficient: −0.1735, *p* < 0.01), the group with symptoms during menstruation alone (regression coefficient: −0.0985, *p* < 0.01), and the group with both severe PMS and symptoms during menstruation (regression coefficient: −0.1537, *p* < 0.01) had significantly lower relative presenteeism than the group without severe PMS or menstrual symptoms. Among all covariates, statistical significance was observed in the regression coefficients for age, overworking hours, and K6 scores (*p* < 0.01).

## 4. Discussion

This study demonstrated that participants with any of the symptoms of severe PMS and both severe PMS and menstrual symptoms had lower presenteeism during the perimenstrual period compared with those with no or slight symptoms. Unlike previous studies showing performance during the presence of menstrual symptoms (absolute absenteeism) [24], the total number of days of perceived decrease in workload and working hours [17], or the proportion of those who experienced a decrease in performance during the period when PMS exists [19], this study illustrated the ratio of work performance during the perimenstrual periods to that during inter-menstrual periods to clarify the extent to which work performance changes throughout the menstrual cycles by severity of symptoms.

Relative presenteeism was 69% (SD: 21) for those with severe PMS alone, and 69% (SD: 27) for those with both severe PMS and symptoms during the menstruation period (*p* < 0.01). These figures are lower than those obtained in a study targeting the general female population in the US [4], which found that work performance decreased to approximately 85% during perimenstrual periods and was also lower than the cutoff (80%) for the relative presenteeism of Japanese workers [37]; this indicates that severe menstrual symptoms are associated with lower presenteeism in this study. By contrast, the presenteeism of Japanese women diagnosed with dysmenorrhea severe enough to require treatment ranged from 39.8% to 48.2% [4], which was lower than the relative presenteeism in the present study. Taken together, relative presenteeism in this study was not too low or too high; thus, the results are reasonable. However, caution should be exercised when comparing the results to other studies, as most previous studies were conducted in other countries or other regions of Japan. For examples, in the case of studies targeting Japanese women [4], recruitment was conducted nationwide to ensure representativeness reflecting age, occupation, income, and other demographic factors by region. Meanwhile, another cross-sectional study has been conducted by us (as yet unpublished) with over 3000 participants, including women working in areas other than central Tokyo, and regional differences were examined. 

Relative presenteeism in those with both severe PMS and symptoms during menstruation was not necessarily the most significant, suggesting that PMS alone has a greater impact on work performance. In this study, the relative presenteeism of those with severe PMS alone and those with both PMS and symptoms during menstruation were almost the same: 69% (SD: 21) and 69% (SD: 27), respectively. The first reason that relative presenteeism in the severe PMS alone group decreased may be the number of respondents; more women in this study had PMS (*n* = 98, 42%) with the sum of the number of those who had any of the severe PMS symptoms (*n* = 49) as well as both severe PMS and symptoms during menstruation (*n* = 44), than those with symptoms during menstruation (*n* = 67, 29%) with the sum of the number of those who have symptoms during menstruation (*n* = 23) as well as both severe PMS and symptoms during menstruation (*n* = 44). Nevertheless, around 40–50% of women may experience PMS regularly [38], while those with menstrual symptoms during menstruation accounted for 45–95% [7], implying that many women, in general, are unaware of PMS. Women in this study had easier access to health information and services because their bases were in central Tokyo; therefore, participants were likely to be aware of and report PMS-related symptoms. To examine the underlying reasons why participants in this study were more likely to report severe PMS than the general population, an analysis using nationally representative data from working women is required. The second reason may be that PMS is not alleviated or is not controlled appropriately and interferes with productivity during menstruation. It is easier for women to take leave or take care of themselves during the period when bleeding is visible and recognizable, while the features of PMS, which are difficult to visualize, make it difficult to recognize the symptoms. Given the above discussion, it is necessary to measure the comprehension level of PMS among women and to disseminate knowledge and techniques to cope with PMS in the future.

The factors that strongly influenced the relationship between menstrual symptoms and relative presenteeism included age, overtime hours, work-self balance, and psychological distress. The results were partially similar to those of a previous study [18], which found low work-life balance, less control over overtime work, and anxiety/depression to be influential. The results of the present study suggest that measures to improve work styles or working environments, such as improving work-life balance and overtime hours, may positively affect performance during menstruation. The present study found the effect of age to be statistically stronger than that reported in a previous study. Previous research has shown that PMS tends to be more severe for women in their 20s and is likely to ease with age and pregnancy [39]. Thus, the fact that 70% of the participants in this study were in their 20s and 30s might have affected the results, owing to the effect of age. However, since psychological distress is also included as a menstrual symptom, it is impossible to examine its causal relationship in a cross-sectional study. Longitudinal studies are required to assess the relationship between psychological distress and menstrual symptoms by measuring anxiety and depression at baseline and end-line performance, as suggested in another study [40]. Previous studies have also identified lifestyle factors, such as diet and exercise, as other relevant factors [41,42]; however, no similar results were obtained in this study. In the future, it will be necessary to comprehensively verify the effects of basic attributes, lifestyle, and work-related factors on the relationship between menstrual symptoms and presenteeism, using methods such as covariance structure analysis that enable the visualization of the strength of the relationship among multiple factors.

### Limitations

This study has several limitations. First, the participants were women working in central Tokyo; their attributes may differ from those of women in other regions. To clarify the relationship between menstrual symptoms and work performance more accurately, an evaluation of women working in other regions or a nationally representative sample is needed. 

Second, participation in this study was on a voluntary basis and the sample size was relatively small. Therefore, the participants were likely to be highly health-conscious with higher health literacy than the general population, thereby making it difficult to generalize the results.

Third, this study assessed the frequency of menstrual symptoms in general, without reference to any time boundary or specific period. The severity of menstrual symptoms varies from cycle to cycle among individuals, and some reporting bias may occur with regard to the point in time that symptoms are reported. Future research is required to elucidate the days of periods when surveying menstrual symptoms.

Fourth, this study assessed the frequency of menstrual symptoms, but did not consider their severity. This is because it was difficult to ask participants to answer a large volume of questionnaires using scales such as the Menstrual Distress Questionnaire (MDQ), which have been verified for their reliability and validity [43], owing to time constraints in responding to the questions. In future research, a simplified version of the MDQ or a scale that can comprehensively evaluate the severity and frequency of menstrual symptoms should be developed. 

## 5. Conclusions

This study demonstrated that severe PMS alone and both severe PMS and symptoms during menstruation particularly decreased work performance compared with during the inter-menstrual period. Therefore, it is crucial to enhance personal skills in order to maintain work performance at any time during the menstrual cycle by taking effective measures at the individual level, as well as improving the working system and environment, as initiated by companies.

## Figures and Tables

**Table 1 ijerph-21-00313-t001:** Demographic characteristics of the 238 eligible participants.

Demographic Variables		*n*	%
Age	Years (Mean ± SD)	35.8 ± 8.0	
	20–29	63	26.5
	30–39	95	39.9
	40–49	68	28.6
	50–59	11	4.6
	60<	1	0.4
Academic background	University grade	198	83.2
	Others	40	16.8
Marital status	Unmarried	144	60.5
	Married	94	39.5
Presence of children	Yes	61	25.6
	No	177	74.4
Time for housework and childcare (hours, Mean ± SD)	Weekday		
	20–29	1.2 ± 0.7	26.5
	30–39	2.7 ± 2.5	39.9
	40–49	2.8 ± 2.2	28.6
	50–59	2.4 ± 1.4	4.6
	60<	2.0 ± NA	0.4
	Weekend		
	20–29	2.1 ± 1.2	26.5
	30–39	5.1 ± 4.7	39.9
	40–49	5.0 ± 4.4	28.6
	50–59	4.0 ± 2.8	4.6
	60<	2.0 ± NA	0.4
Occupation	Clerical job	152	63.9
	Service job	10	4.2
	Professional job	46	19.3
	Sales job	28	11.8
	Others	2	0.8
Position *	Management	22	9.9
	Non-management	199	89.6
	Executive position	1	0.5
Frequently severe PMS or symptoms during menstruation		Variables	%
Group without severe PMS or menstrual symptom		119	50.0
Group with any of the symptoms of severe PMS alone		50	21.0
Group with any of the severe symptoms during menstruation alone		23	9.7
Group with both severe PMS and symptoms during menstruation		46	19.3
Work-related factors		*n*	%
Friendly atmosphere in workplace *	Yes	192	86.5
	No	30	12.6
Job self-control (Score ranges from 3 to 12, Mean ± SD)		8.9 ± 1.8	
Work engagement *	Yes	105	47.3
	No	117	52.7
Work-self balance (Score ranges from 4 to 16, Mean ± SD)		10.6 ± 2.5	
Easy leave *	Yes	203	91.4
	No	19	8.6
Flexi-time or short-working-hour system *	Yes	185	83.3
	No	37	16.7
Overworking hours per day *	<1	50	22.5
	1–3	130	58.6
	3–5	30	13.5
	5<	12	5.4
Health-related factors		*n*	%
General health	Good	3	1.3
	Fairly good	50	21.0
	Moderate	77	32.4
	Not so good	77	32.4
	Bad	31	13.0
K6 score (Score from 0 to 24, Mean ± SD)		5.1 ± 4.9	
Experience of seeing obstetricians and gynecologists	Yes	72	30.3
	No	166	69.7
Literacy on symptoms during menstruation	Knowledge of symptoms and coping approach	25	10.5
	Others	213	89.5
Literacy on PMS	Knowledge of symptoms and coping approach	60	25.2
	Others	178	74.8
Consultation on women’s health issues	Yes	201	84.5
	No	37	15.5

* Missing value was 16; thus, the total sample size was 222.

**Table 2 ijerph-21-00313-t002:** The association between the existence of frequently severe PMS or symptoms during menstruation and relative presenteeism (ANOVA).

Existence of Frequently Severe Menstrual Symptoms	*n*	Relative Presenteeism × 100 (%) ***	
		Mean	SD
All participants	232	81	25
		*p* value	<0.01 *
Group without severe PMS or menstrual symptoms (1)	116	91	23
Group with any of the symptoms of severe PMS alone (2)	49	69	21
Group with any of the severe symptoms during menstruation alone (3)	23	76	16
Group with both severe PMS and symptoms during menstruation (4)	44	69	27
Tukey test **		Difference	*p*-adjusted
(2)-(1)		−22	<0.01
(3)-(1)		−16	0.02
(4)-(1)		−23	<0.01

* *p* value obtained from analysis of variance to measure the difference in relative presenteeism among the four groups ((1)-(4)); ** The results of Tukey test to measure the difference in relative presenteeism among every two groups among the four groups. *** Relative presenteeism ×100 was calculated by dividing performance during perimenstrual periods by presenteeism over the past 1 month, multiplied by 100.

**Table 3 ijerph-21-00313-t003:** Linear regression of relative presenteeism on the existence of frequently severe PMS or symptoms during menstruation *.

	Relative Presenteeism							
	Model 1 (*n* = 238)		Model 2 (*n* = 238)		Model 3 (*n* = 216)		Model 4 (*n* = 216)	
	Regression Coefficient	*p* Value	Regression Coefficient	*p* Value	Regression Coefficient	*p* Value	Regression Coefficient	*p* Value
Frequently severe PMS or symptoms during menstruation **								
Group without severe PMS or menstrual symptoms	ref	ref	ref	ref	ref	ref	ref	ref
Group with any of the symptoms of severe PMS alone	−0.2234	<0.01	−0.2126	<0.01	−0.1984	<0.01	−0.1735	<0.01
Group with symptoms during menstruation alone	−0.1578	<0.01	−0.1356	0.01	−0.1244	<0.01	−0.0985	<0.01
Group with both severe PMS and symptoms during menstruation	−0.2254	<0.01	−0.2095	<0.01	−0.2015	<0.01	−0.1537	<0.01
Demographic factors								
Age			0.0040	0.06	0.0052	0.02	0.0058	<0.01
Academic background			0.0487	0.25	0.0410	0.38	0.0544	0.28
Marital status			−0.0030	0.94	−0.0065	0.88	−0.0180	0.71
Presence of children			0.0198	0.68	0.0202	0.69	0.0325	0.53
Work-related factors								
Friendly atmosphere in workplace					0.0295	0.55	−0.0004	0.46
Job control					−0.0151	0.17	−0.0171	0.26
Work engagement					−0.0252	0.53	−0.0396	0.29
Work-self balance					0.0200	0.02	0.0131	0.32
Easy leave					0.1063	0.08	0.1070	0.14
Flexi-time or short-working-hour system					−0.0020	0.97	−0.0271	0.97
Overworking hours per day					0.0459	0.05	0.0513	0.05
Health-related factors								
General health							−0.0023	0.80
K6							−0.0463	<0.01
Literacy on symptoms during menstruation							−0.0229	0.36
Literacy on PMS							−0.0060	0.85
Consultation on women’s health issues							0.0160	0.75
Adjusted R2 and *p* value	0.1705	<0.01	0.1749	<0.01	0.1937	<0.01	0.2058	<0.01

* Multi-regression analyses were conducted to examine the association between frequently severe PMS or symptoms during menstruation and relative presenteeism; ** group without severe PMS or menstrual symptoms was used as a reference to compare the impact of PMS alone, symptoms during menstruation alone, and both PMS and symptoms during menstruation.

## Data Availability

Data for this study cannot be publicly shared due to privacy and ethical constraints. We are committed to transparency and have provided a detailed methodology in the article to support reproducibility. For specific inquiries, please contact the corresponding author.

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
