# Peer review of "The Association between Menstrual Symptoms and Presenteeism: A Cross-Sectional Study for Women Working in Central Tokyo"

_ijerph, 2024, doi:10.3390/ijerph21030313_

Round 1

Reviewer 1 Report

Comments and Suggestions for Authors

Review. Please, find my comments below and also from PDF

Lines 21 and 22

“The severity of PMS and symptoms during menstruation were categorized based on the frequency…”

I was looking forward to frequency during or within which time limit.

Line 24

An analysis of variance..”

 Maybe authors could add (ANOVA)

 Lines 28 – 29

 “ ….were 91% (standard deviation (SD) 23), 69% 28 (SD 21), 76% (SD 16), and 69% (SD 27), respectively…...

 I did not catch those figures directly, but I understood those when I checked the result section. Authors could clarify a bit how those relate to relative presenteeism. Should it be as follows: Participants were divided into four groups; 1) None/mild, 2) PMS alone, 3) Symptoms during menstruation alone, and 4) Both PMS and symptoms during menstruation, and mean relative presenteeism were 91% (standard deviation (SD) 23), 69% 28 (SD 21), 76% (SD 16), and 69% (SD 27), respectively (p<0.01).

 Line 34

 What about keyword presenteeism. It was not at a list.

 Lines 38- 39

 “…considerably lower women’s health-related quality of life and work performance.”

 One reference after a sentence would be great.

 Lines 47- 48

 “ Recently, work performance and productivity have gained attention in business settings.”

 Yes, you are right, but you could add e.g., one relevant reference after a sentence.

 Lines 51-52

 “ ….symptoms are 4,333 USD per year in the US..”

 Please, clarify this USD figure. Is it per woman per year or something else?

 Lines 55-59

 “ Hence, from the perspective of both corporate performance and the national economy, it is imperative to prevent absenteeism and poor presenteeism. One of the most vital issues in the field of occupational health is maintaining work performance at any time during the menstrual cycle by taking effective measures at appropriate times to prevent or alleviate menstrual symptoms.”

 I am suggesting moving that part of text to discussion section or to add some references or to add statement that this is authors opinion.

 Line 60

 “ Most previous studies….”

 Please, add some relevant references as examples.

 Line 74

 “…relative presenteeism, which….”

 You could add a reference to relative presenteeism. For example, https://link.springer.com/article/10.1007/s41105-021-00339-4

 Line 81

 In previous studies,..”

 Please, add some reference, which previous studies.

 Line 90

 “ women working in central Tokyo.”

 I am curious to know why authors indicate that women in central Tokyo instead of Japan. Are there some business or career issues that may differ between central Tokyo and other parts of Japan. If there are, the authors may write line or two.

 Line 98

 the survey was conducted…”

 Please, clarify a language for survey. I guess Japanese or was it available also in English.

 Line 107

 “2.2. Study Population”

 In my opinion, you could mention already here how many participants you had (312) and if possible, to estimate what is the relative share that they represent regarding eligible women in nine target companies. Please, also clarify if all of them were Japanese or if population existed also foreigners.

 Line 123

 “ Participants were asked regarding the frequency…”

 Please, clarify a bit what was a time scale for assessing frequency? Was is during a year or something else?

 Line 142 – 143

 “Relative presenteeism is the ratio of actual performance to work performance of most coworkers.”

 Please, add a reference.

 Line 144

 “restricting the range of ration to 0.25–2..”

 Maybe I did not catch this range of ration. Should it be range of ratio and max of ratio 1?

 Line 152

 “the Job Stress Questionnaire..”

 Maybe a reference for that would be good.

 Line 169

 “ the Kessler Psychological Distress Scale (K6)..”

 Please, add a reference to that scale.

 Line 237 – 238 in Table 2

 “ *** Relative presenteeism×100 was calculated by dividing presenteeism during perimenstrual periods by presenteeism over the past 1 month, multiplied by 100.”

According to definition above, relative presenteeism is the ratio of performance during the perimenstrual period to that during the intra-menstrual period, but it the table authors are using presenteeism instead of performance. Please, clarify that issue.

Line 257

 had poor presenteeism during…”

 In my opinion presenteeism is always poor and a productivity robber. You could define shortly what does poor presenteeism mean.

 Line 263 – 264

 “ this indicates that severe menstrual symptoms are associated with lower presenteeism.”

 You could add that “in this study”. It is difficult to generalize the result to cover severe menstrual symptoms in every population. You mentioned that about 50% of participants claimed severe symptoms in either PMS or during menstruation, but did you mention somewhere the prevalence of presenteeism in the studied population. You could add it in discussion and abstract.

 Line 312

 Previous studies have also…”

 Please, add a reference.

Reviewer 2 Report

Comments and Suggestions for Authors

Okamoto et al managed to investigate the impact of menstrual symptoms on presenteeism. I have the following comments: 

1: The study population was divided into one, PMS alone, Menstrual symptoms alone, and PMS and menstrual symptoms). The authors may consider adding a new category (Any), including those with any PMS alone, Menstrual symptoms alone, and PMS and menstrual symptoms altogether.

2: The authors may consider stratifying the presenteeism ratio to transfer the continuous variable into a category and then do a logistic regression analysis. This analysis approach may give more insight into the effect of menstrual symptoms on presenteeism. The authors can conclude that premenstrual symptoms reduced the odds of presenteeism by X%. This will be easier to understand for policymakers.

3: The cross-sectional design is a limitation in itself. It is presumptive to assume that menstrual symptoms are the reasons behind low presenteeism because both variables were assessed at the same time. Thus, it is advised to use the term (association between menstrual symptoms and presenteeism). 

4: Some confounders, such as stress and sleep, might have affected the results. These factors should be admitted in the limitation section.

5: PMS and menstrual symptoms were assessed as a whole. PMS symptoms were not assessed individually. This is another limitation.

6: PMS pain and symptoms vary across cycles. Women might have reported the worst cycles. This is a potential bias.

7: The sample size is small. It should be justified.

8: The response rate and reasons behind non-response should be clarified.

9: Several studies regarding work absenteeism in Japan were not mentioned.

Comments on the Quality of English Language

Moderate editing.

Reviewer 3 Report

Comments and Suggestions for Authors

The idea of the study is relevant and meaningful, but the manuscript is not enough good quality to publish in present form. Below you will find my comments which justify such my opinion.

The problem is not clearly defined in the Introduction. The theoretical background of the study is low quality. The authors should describe the concepts which they use in the study, especially those which are interpreted differently by different authors, for example, presenteeism. Also, the authors should show what is the interaction between the variables which are measured. The relationship between context variables (the authors call them covariates) and two main variables should be explained also. There are no hypotheses and the theoretical model of the study in the manuscript.

The methods and instruments of the measurement should be described more comprehensively. Now the description does not allow to understand which instrument was used to evaluate Relative presenteeism and what does it mean "revised scale". The same comment is regarding measurement instruments of other variables (covariates).

According previous comments it is difficult to evaluate how much the Discussion and Conclusions are validated/justified.

Reviewer 4 Report

Comments and Suggestions for Authors

Introduction

Line 47, more explanation with references on what is presenteeism, this is a not so well understood concept so need to be very well explained in the beginning. Has presenteeism been studied due to other clinical or health conditions based on available literature?

Line 47-59 needs better references and explanation of what other research(ers) in the field have concluded.

Line 75 “performing the same job, is commonly used in the fields of management and occupational health” this sentence needs reference and explanation.

Overall, introduction should be improved. This paper is studying the effect on menstrual symptoms on work performance. Two things have to be better explained to lay the foundation: 1) what is presenteeism and how is it different from absenteeism; 2) What this study aims to add to the information already available, for example is it the definition of PMS, Menstrual cycle and intermenstrual cycle etc or is it the effect of these on work performance?.

Methods

Line 144 “restricting the range of ration to 0.25–2 [10-12]. “ it is not clear what is meant by this sentence.

Sentence 144 to 147 is not clearly explained and reading the references does not give a clear picture. This has to be better explained.

2.3.2. Relative Presenteeism should be explained in more detail so that readers can reproduce the work if they wanted to.

Discussion

Possible explanations as to why there is differences between this study results and other countries and other studies in Japan should be given. This will help readers to derive better conclusions. The authors have compared their results with other studies; however no mention has been made of any intervention by any researchers or institutions to use the results of studies pertaining to presenteeism and menstrual symptoms. This is an important discussion that needs to be provided, otherwise what is the application of this and similar studies to help the workers?

References

Many references are in the acceptable format, some doi are not correct, need to be checked

Comments on the Quality of English Language

There are some minor English errors that need to be checked

Round 2

Reviewer 2 Report

Comments and Suggestions for Authors

Except for minor grammatical errors that should be considered, I have no more comments.

Comments on the Quality of English Language

Minor corrections are needed.

Reviewer 3 Report

Comments and Suggestions for Authors

Dear Authors,

thank you for efforts to do your manuscript of a better quality. I have no comments.

Reviewer 4 Report

Comments and Suggestions for Authors

Thank you for addressing all the comments